# Consolidation alters motor sequence-specific distributed representations

Basile Pinsard[1,2]*, Arnaud Boutin[2,3], Ella Gabitov[2,3], Ovidiu Lungu[2], Habib Benali[1,4], Julien Doyon[2,3,5]*

[1]Laboratoire d'Imagerie Biomédicale, Sorbonne Université, CNRS, INSERM, Paris, France; [2]Functional Neuroimaging Unit, Centre de Recherche de l'Institut Universitaire de Gériatrie de Montréal, Montreal, Canada; [3]McConnell Brain Imaging Centre, Montreal Neurological Institute, McGill University, Montreal, Canada; [4]PERFORM Centre, Concordia University, Montreal, Canada; [5]Department of Neurology and Neurosurgery, Montreal Neurological Institute, McGill University, Montreal, Canada

**Abstract** Functional magnetic resonance imaging (fMRI) studies investigating the acquisition of sequential motor skills in humans have revealed learning-related functional reorganizations of the cortico-striatal and cortico-cerebellar motor systems accompanied with an initial hippocampal contribution. Yet, the functional significance of these activity-level changes remains ambiguous as they convey the evolution of both sequence-specific knowledge and unspecific task ability. Moreover, these changes do not specifically assess the occurrence of learning-related plasticity. To address these issues, we investigated local circuits tuning to sequence-specific information using multivariate distances between patterns evoked by consolidated or newly acquired motor sequences production. The results reveal that representations in dorsolateral striatum, prefrontal and secondary motor cortices are greater when executing consolidated sequences than untrained ones. By contrast, sequence representations in the hippocampus and dorsomedial striatum becomes less engaged. Our findings show, for the first time in humans, that complementary sequence-specific motor representations evolve distinctively during critical phases of skill acquisition and consolidation.
DOI: https://doi.org/10.7554/eLife.39324.001

*For correspondence:
basile.pinsard@gmail.com (BP);
julien.doyon@mcgill.ca (JD)

**Competing interests:** The authors declare that no competing interests exist.

## Introduction

Animals and humans are able to acquire and automatize new sequences of movements, hence allowing them to expand and update their repertoire of complex goal-oriented motor actions for long-term use. To investigate the mechanisms underlying this type of procedural memory in humans, a large body of behavioral studies has used motor sequence learning (MSL) tasks designed to test the ability to perform temporally ordered and coordinated movements, learned either implicitly or explicitly and has assessed their performances in different phases of the acquisition process (*Korman et al., 2003*; *Abrahamse et al., 2013*; *Diedrichsen and Kornysheva, 2015*; *Verwey et al., 2015*). While practice of an explicit MSL task leads to substantial within-session execution improvements, there is now ample evidence indicating that between-session maintenance of, and even increases in, performance can be observed after a night of sleep (*Landry et al., 2016*; *Nettersheim et al., 2015*), while performance is unstable and tends to decay during an equal period of wakefulness (*Doyon et al., 2009a*; *Brawn et al., 2010*; *Nettersheim et al., 2015*; *Landry et al., 2016*). Therefore, it is thought that sleep favors reprocessing of the motor memory trace, thus promoting its consolidation for long-term skill proficiency (*Fischer et al., 2002*, see *King et al., 2017*; *Doyon et al., 2018* for recent in-depth reviews).

Functional magnetic resonance imaging (fMRI) studies using General-Linear-Model (GLM) contrasts of activation have also revealed that MSL is associated with the recruitment of an extended network of cerebral (*Hardwick et al., 2013*), cerebellar and spinal regions (*Vahdat et al., 2015a*), whose contributions differentiate as learning progresses (*Karni et al., 1998*; *Dayan and Cohen, 2011*; *Doyon et al., 2018*). In fact, plastic changes (*Ungerleider et al., 2002*; *Doyon and Benali, 2005*) are known to occur within the initial training session, as well as during the offline consolidation phase, the latter being characterized by a functional 'reorganization' of the nervous system structures supporting this type of procedural memory function (*Rasch and Born, 2008*; *Born and Wilhelm, 2012*; *Albouy et al., 2013b*; *Dudai et al., 2015*; *Bassett et al., 2015*; *Fogel et al., 2017*; *Vahdat et al., 2017b*). More specifically, MSL is known to activate a cortical, associative striatal and cerebellar motor network, which is assisted by the hippocampus during the initial 'fast-learning' phase (*Albouy et al., 2013b*). Yet, when approaching asymptotic behavioral performance after longer periods of practice, activity within the hippocampus and cerebellum decreases while activity within the sensorimotor striatum increases (*Doyon et al., 2002*), both effects conveying the transition to the 'slow-learning' phase. Other studies have then shown that the same striatal regions are reactivated during sleep spindles (*Fogel et al., 2017*), hence contributing to the progressive emergence of a reorganized network (*Debas et al., 2010*; *Vahdat et al., 2017b*) that is further stabilized when additional MSL practice is extended over multiple days and weeks (*Lehéricy et al., 2005*).

A critical issue typically overlooked by previous MSL neuroimaging research using GLM-based activation contrasts, however, is that learning-related changes in brain activity do reflect the temporal evolution of recruited processes during blocks of practice, only some of which may be specifically related to brain plasticity induced by MSL. For instance, increases in activity could not only signal a greater implication of the circuits specialized in movement sequential learning per se, but could also result from the inherent faster execution of the motor task. Likewise, a decrease in activity could either indicate some form of optimization and greater efficiency of the circuits involved in executing the task (*Wu et al., 2004*), or could show the reduced recruitment of non-specific networks supporting the acquisition process. Therefore, even with the use of control conditions to dissociate sequence-specific from non-specific processes (*Orban et al., 2010*), the observed large-scale activation differences associated with different learning phases do not necessarily provide direct evidence of plasticity related to the processing of a motor sequence-specific representation (*Berlot et al., 2018*). Furthermore, it is also conceivable that these plastic changes could even occur locally without significant changes in the GLM-based regional activity level. Finally, in most studies investigating the neural substrate mediating the consolidation process of explicit MSL, the neural changes associated with this mnemonic mechanism are assessed by contrasting the brain activity level of novice participants between their initial training and a delayed practice session. Therefore, they measure not only plasticity for sequence-specific (e.g. optimized chunks), but also task-related competence (e.g. habituation to experimental apparatus, optimized execution strategies, attentional processes). The latter proficiency is notably observed when participants practice two motor sequences in succession and the initial performance during sequence execution is significantly better for the subsequent than for the first sequence.

To address these specificity limitations, multivariate pattern analysis (MVPA) has been proposed to evaluate how local patterns of activity are able to reliably discriminate between stimuli or evoked memories of the same type over repeated occurrences, hence allowing to test information-based hypotheses that GLM contrasts cannot inquire (*Hebart and Baker, 2018*). In the MSL literature, only a few studies have used such MVPA approaches to identify the regions that specialize in processing the representation of learned motor sequences (*Wiestler and Diedrichsen, 2013*; *Nambu et al., 2015*; *Wiestler et al., 2011*; *Kornysheva and Diedrichsen, 2014*; *Yokoi et al., 2018*). These studies, however, mainly focused on extensively practiced sequences over multiple training sessions across multiple days. For instance, in a recent study covering dorsal cerebral cortices only (*Wiestler and Diedrichsen, 2013*), cross-validated classification accuracy was measured separately on activity patterns evoked by the practice of trained and untrained sets of sequences. The authors showed that the extended training increased sequence discriminability in a network spanning bilaterally the primary and secondary motor as well as parietal cortices. In another study (*Nambu et al., 2015*) that aimed to analyze separately the preparation and execution of sequential movements, representations of extensively trained sequences were identified in the contralateral dorsal premotor and supplementary motor cortices during preparation, while representations related to the

execution were found in the parietal cortex ipsilaterally, the premotor and motor cortices bilaterally as well as the cerebellum. In both studies, the regions carrying sequence-specific representations overlapped only partly with those identified using GLM-based measures, hence illustrating the fact that coarser differences in activation between novel and trained sequences does not necessarily provide evidence of plasticity for sequential information. However, the classification-based measures they used may have biased their parametric statistical results by violating both the normality assumption and theoretical null-distribution (*Jamalabadi et al., 2016*; *Allefeld et al., 2016*; *Combrisson and Jerbi, 2015*; *Varoquaux, 2018*) and may have thus been suboptimal in detecting representational changes (*Walther et al., 2016*).

As a part of a larger research program interested in identifying the neural substrates implicated in both consolidation and reconsolidation processes, the present study aimed to address both the critical issues overlooked by previous research investigating the early phases of MSL consolidation with GLM-based approach described above, as well as the limitations encountered when using classifier-based MVPA methods. Specifically, we employed a recently developed MVPA approach (*Nili et al., 2014*) that is unbiased and more sensitive to continuous representational changes (*Walther et al., 2016*), such as those that occur in the early stage of MSL and consolidation (*Albouy et al., 2013c*). Our experimental manipulation allowed us to isolate sequence-specific plasticity, by extracting patterns evoked through practice of both consolidated and new sequences at the same level of proficiency in the task and by computing this novel multivariate distance metric using a searchlight approach over the whole brain in order to cover cortical and subcortical regions critical to MSL. Based on theoretical models (*Albouy et al., 2013b*; *Doyon et al., 2018*) derived from imaging work in humans and invasive animal studies, we hypothesized that offline consolidation following training would induce greater cortical and striatal as well as weaker hippocampal sequence-specific representations.

## Results

To investigate changes in the neural representations of motor sequences occurring during learning, young, healthy participants (n = 18) practiced two five-element sequences of finger movements (executed through button presses) separately on two consecutive days. On the third day, participants were required to execute again the same two sequences, then considered to be consolidated, together with two new five-element untrained sequences. This practice session consisted of 64 pseudo-randomly ordered short blocks split in two runs, with 16 blocks of each sequence. All four sequences were executed using their non-dominant left hand while functional MRI data was acquired.

### Behavioral performance

We analyzed the behavioral performance related to the four different sequences using a repeated-measure mixed-effects model (with grouping within subject). First, we tested for differences in performances between new and consolidated sequences performances (i.e., speed as average duration of correct sequences and number of errors per block), by taking as fixed-effect the learning stage (new or consolidated) and number of blocks and their interaction, and as random effect the sequences and blocks to account for between-subject differences in performance baseline and rate of change (See *Supplementary files 1* and *2* respectively for speed and errors results). As expected, new sequences were performed more slowly ($\beta = .365, SE = 0.047, p < .001$) and less accurately ($\beta = -0.304, SE = 0.101, p < .001$) than the consolidated ones. Significant improvement across blocks was observed for new sequences as compared with consolidated sequences in term of change of speed ($\beta = -0.018, SE = 0.002, p < .001$), thus showing an expected learning curve visible in *Figure 1*. Yet accuracy did not show significant improvement ($\beta = 0.014, SE = 0.010, p = 0.152$), likely explained by the limited precision of this measure that ranges discretely from 0 to 5. By contrast, the consolidated sequences did not show significant changes in speed ($\beta = -0.006, SE = 0.005, p = 0.192$) nor accuracy ($\beta = -0.006, SE = 0.057, p = 0.919$), the asymptotic performances being already reached through practice and the consolidation process. While most studies in the field have measured retention and even offline gains between the end of training and the beginning of a post-sleep retest, we chose here to measure consolidation as the difference at the same point in time between the new and consolidated sequences. We also measured the difference in speed (mean correct sequence

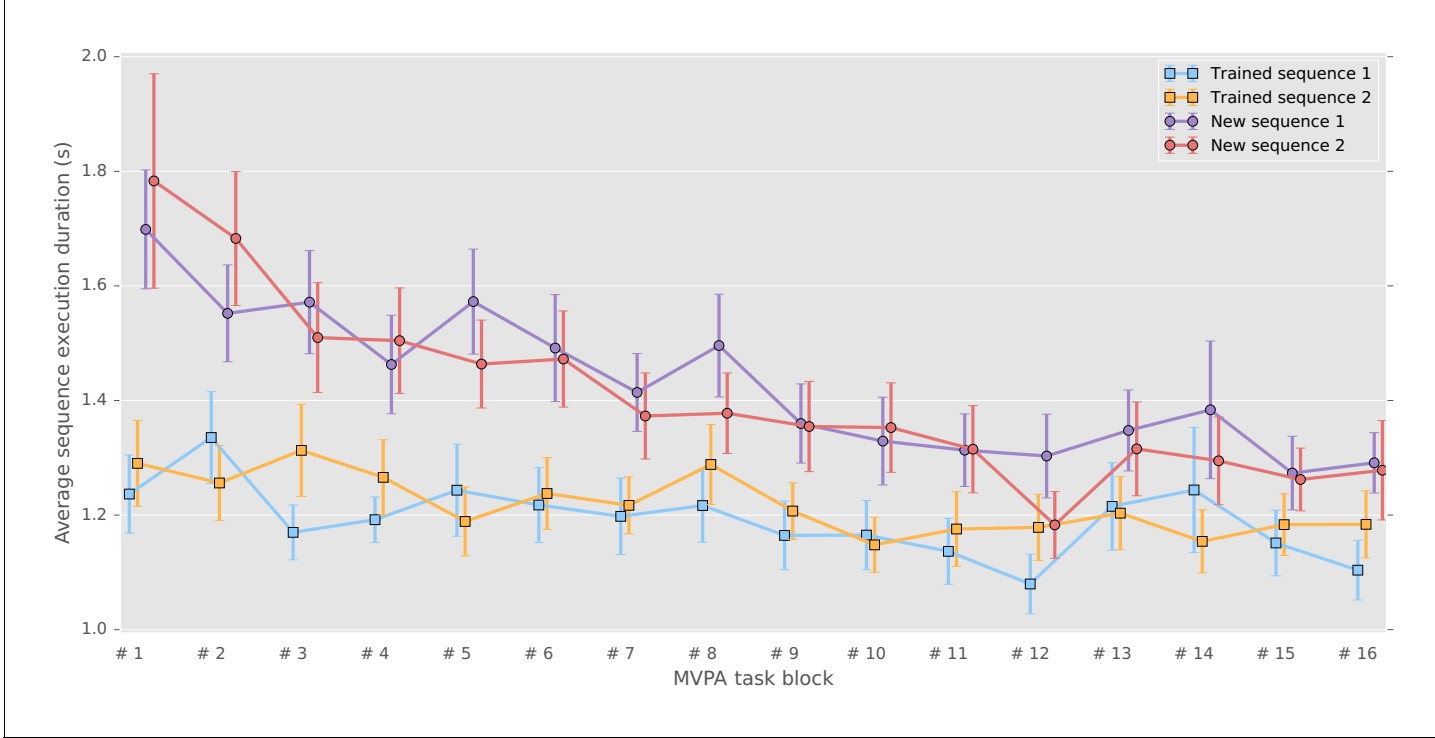

**Figure 1.** Correct sequence durations (average and standard error of the mean across participants) across the MVPA task blocks.
DOI: https://doi.org/10.7554/eLife.39324.002
The following figure supplement is available for figure 1:

**Figure supplement 1.** Sequence duration of trained sequences in the first blocks for each sequence on the last experimental day (mean and standard error across participants).
DOI: https://doi.org/10.7554/eLife.39324.003

duration) between the last two blocks of training and the first two blocks of retest for the consolidated sequences (fixed-effect:post, random-effect:sequence) in order to be comparable to previous studies, and found a significantly lower sequence duration in the beginning of the retest ($\beta = 0.086, SE = 0.023, p = .000141$).

Second, we tested separately the new and consolidated sequences to verify that they did not show consistent differences across subjects using a model that combined sequence, blocks and their interaction as fixed effect (model and results can be found in *Supplementary files 3* and *4* respectively). Importantly, there were also no significant differences between the two consolidated sequences in term of speed ($\beta = 0.031, SE = 0.026, p = 0.234$) and accuracy ($\beta = -0.030, SE = 0.111, p = 0.789$), nor between the two new sequences with respect to speed ($\beta = 0.025, SE = 0.045, p = 0.577$) and accuracy ($\beta = -0.245, SE = 0.138, p = 0.076$). To verify that behavioral differences could not be accounted by the retest of trained sequences (7 blocks of 12 repetitions for each sequence) preceding the task analyzed here, we compared the sequence duration of the five first sequences of the retest blocks with the duration of the untrained sequences during the present task. The values reported in supplementary material (*Figure 1—figure supplement 1*), show that the trained sequences are still performed faster than the untrained sequences ($\beta = -.431, SE = 0.090, p < 10^{-5}$).

## A common distributed sequence representation irrespective of learning stage

From the preprocessed functional MRI data we extracted patterns of activity for each block of practice, and computed a cross-validated Mahalanobis distance (*Nili et al., 2014*; *Walther et al., 2016*) using a Searchlight approach (*Kriegeskorte et al., 2006*) over brain cortical surfaces and subcortical regions of interest. An illustration of the method can be found in *Figure 2—figure supplement 4*.

Such multivariate distances, when positive, demonstrate that there is a stable difference in activity patterns between the conditions compared, and thus reflect the level of discriminability between these conditions. To assess true representational patterns, and not mere global activity differences, we computed this discriminability measure for sequences that were at the same stage of learning, thus separately for consolidated and new sequences. From the individual discriminability maps, we then measured the prevalence of discriminability at the group level, using non-parametric testing with a Threshold-Free-Cluster-Enhancement approach (TFCE) (*Smith and Nichols, 2009*) to enable locally adaptive cluster correction.

To extract the brain regions that show discriminative activity patterns for specific sequence during both learning stages, we then submitted these separate group results for the consolidated and new sequences to a minimum-statistic conjunction analysis. A large distributed set of regions (*Figure 2*) displayed significant discriminability, including the primary visual, as well as the posterior parietal, primary and supplementary motor, premotor and dorsolateral prefrontal cortices. (See the statistical maps for each learning stage separately in the Supplementary material (*Figure 2—figure supplement 1*, *Figure 2—figure supplement 2*).

To complement this result we extracted the main effect of activation during this task (*Figure 2—figure supplement 3*), showing that pattern distances do not exactly correspond to the regions significantly activated in mass-univariate GLM analysis.

## Reorganization of the distributed sequence representation after memory consolidation

In order to evaluate the reorganization of sequence representation undergone by consolidation at the group level, the consolidated and new sequence discriminability maps from all participants were submitted to a non-parametric two-tailed pairwise t-test with TFCE. To ascertain that a greater discriminability in one stage versus the other was supported by a significant level of discriminability within that stage, we then calculated the minimum statistic conjunction (*Nichols et al., 2005*) of the contrast maps with the consolidated and new sequences group results, respectively with the positive and negative contrast differences (*Figure 3*). This procedure equals to mask the contrast with significantly positive simple effects. The resulting map thus represents the subset of voxels that fulfill the union of two logical conjunctions: ((Consolidated>New)∩(Consolidated>0))∪((New>Consolidated)∩ (New>0)) with each sub-test being assessed by thresholding the p-value of individual non-parametric permutation tests.

Discriminability between the consolidated sequences was significantly higher than that between the new sequences in bilateral sensorimotor putamen, thalamus and anterior insula, as well as in the ipsilateral cerebellar lobule IX, posterior cingulate and parietal cortices, and contralaterally in the lateral and dorsal premotor, supplementary motor, frontopolar and dorsolateral prefrontal cortices in addition to cerebellar Crus I. By contrast, the pattern dissimilarity was higher for the new sequences in bilateral hippocampi as well as the body of the caudate nuclei, subthalamic nuclei, and cerebellar Crus II ipsilaterally. Although striatal activity patterns differentiating newly acquired sequences were found in contralateral putamen (*Figure 2—figure supplement 1*), this discriminability was significantly larger for consolidated sequences in sensorimotor regions of the putamen bilaterally.

We also conducted a univariate GLM analysis to contrast the consolidated and new sequences, but no results survived multiple comparison correction. This negative result could be accounted for by the absence of smoothing of signals during preprocessing.

## Discussion

In the present study, we aimed to identify the brain regions whose activity patterns differentiate between representations of multiple motor sequences during their execution in different acquisition phases: newly learned and consolidated. Using an MVPA approach, we considered that stable local patterns of activity could be used as a proxy for the specialization of neuronal circuits supportive of the efficient retrieval and expression of sequential motor memory traces. To investigate the differential pattern strength, we computed the novel unbiased multivariate distance and applied robust permutation-based statistics with adaptive cluster correction.

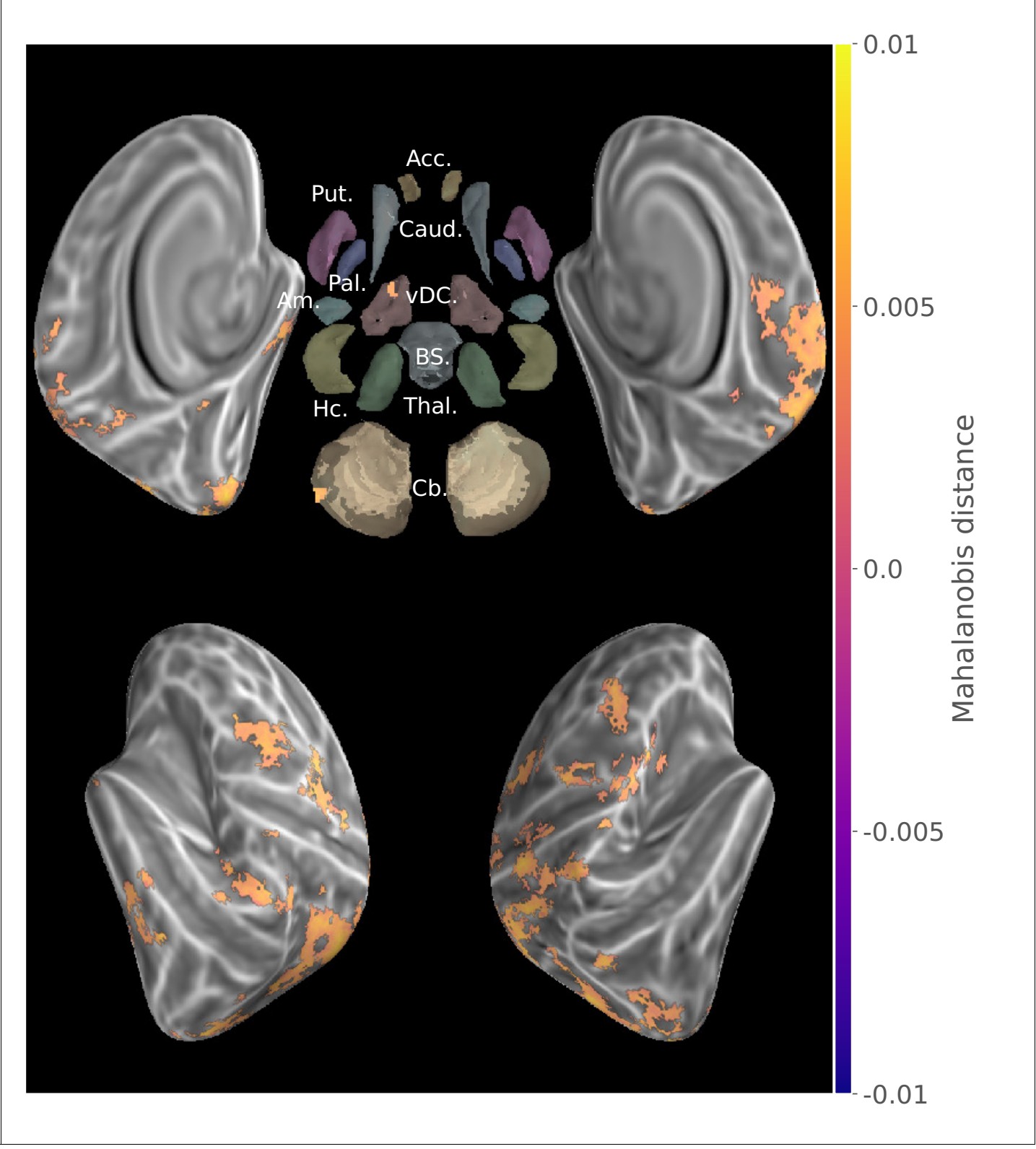

**Figure 2.** Group searchlight conjunction of new and consolidated sequences discriminability maps (z-score thresholded at p < .05 TFCE-cluster-corrected) showing a large distributed set of cortical regions showing sequence disciminative patterns at both learning stages. Regions of interest with Freesurfer colors: Acc.:Accumbens; Put.:Putamen; Caud.:Caudate; Pal.:Pallidum; vDC:ventral diencephalon; Am.:Amygdala; Hc.:Hippocampus; Thal.: Thalamus; Cb.:Cerebellum; BS:brain-stem.

*Figure 2 continued on next page*

*Figure 2 continued*

DOI: https://doi.org/10.7554/eLife.39324.004

The following figure supplements are available for figure 2:

**Figure supplement 1.** Group searchlight map of cross-validated Mahalanobis distance between the two new sequences (z-score thresholded at p < .05 TFCE-cluster-corrected).

DOI: https://doi.org/10.7554/eLife.39324.005

**Figure supplement 2.** Group searchlight map of cross-validated Mahalanobis distance between the two consolidated sequences (z-score thresholded at p < .05 TFCE-cluster-corrected).

DOI: https://doi.org/10.7554/eLife.39324.006

**Figure supplement 3.** Main effect of motor sequence execution during MVPA task.

DOI: https://doi.org/10.7554/eLife.39324.007

**Figure supplement 4.** Illustration of the method: for each neighborhood centered on a gray-matter coordinate, we computed cross-validated Mahalanobis distance matrices.

DOI: https://doi.org/10.7554/eLife.39324.008

## A distributed representation of finger motor sequence

Our results provide evidence for an extended set of brain regions that shows reliable discrimination of sequence-specific activity patterns for both the consolidated and novel sequences. At the cortical level, it encompasses the medial and lateral premotor areas as well as posterior parietal cortices bilaterally and contralateral somatosensory motor cortex. These findings are consistent with earlier MVPA investigations (*Wiestler and Diedrichsen, 2013*; *Nambu et al., 2015*). Indeed, similar discriminative power of motor sequence representations within the ipsilateral premotor and parietal cortices has previously been described (*Wiestler and Diedrichsen, 2013*; *Waters-Metenier et al., 2014*; *Waters et al., 2017*), notably when the non-dominant hand is used for fine dexterous manual skills. Interestingly, we also found significant neural representations for both learning stages in the contralateral primary motor and somatosensory (M1/S1) cortices, more specifically around the hand knob area (*Yousry et al., 1997*) for which finger somatotopy is measurable using fMRI (*Ejaz et al., 2015*). The latter results suggest that these primary cortical regions contribute to the acquisition of experience-related motor sequence memory traces. Yet such an interpretation must be taken with caution, as it has recently been reported that the capacity to discriminate between sequences based upon signals from these regions could simply be due to the stronger activity evoked by the first finger press in the sequence, and not to activity from the whole finger sequence (*Yokoi et al., 2018*). Yet although conjectural, we do not believe that such an effect can explain our pattern of results because, while the newly learned sequences began with different fingers, both consolidated sequences were discriminated despite the fact that the first finger presses were the same. Finally, while being located around the hand knob, the spatial extent of the M1/S1 representation in our study was smaller compared with that found by *Wiestler and Diedrichsen (2013)*. This may be due, however, to differences in our design, notably in the uninterrupted repetition of the motor sequence during practice, and in the fact that none of our sequences engaged the thumb, which has a more distinctive M1/S1 cortical representation than the individual fingers (*Ejaz et al., 2015*). As other regions are expected to show finger somatotopic organization (*Wiestler et al., 2011*), this first finger effect could explain the significant results observed in the map of new sequence discrimination (*Figure 2—figure supplement 1*), such as in a large extent of the dorsal ipsilateral cerebellum.

The conjunction of new and consolidated sequences discriminability maps further revealed that a common cortical processing network, including non-motor primary and associative regions, carries sequential information across learning stages, which can originate from visually presented instruction and short-term memory to motor sequence production. Herein, the visual occipital cortices, which could reflect the processing of the visual stimuli as low-level visual mapping of shapes (*Pilgramm et al., 2016*; *Miyawaki et al., 2008*), as well as the ventro-temporal regions, known to support higher level Arabic number representation (*Shum et al., 2013*; *Peters et al., 2015*), were found to discriminate between sequences in both stages of learning (*Figure 2*). The dorsolateral prefrontal cortex, which also exhibited pattern discriminability, was suggested previously to process the sequence spatial information in working memory, preceding motor command (*Robertson et al., 2001*). In fact, we believe that the cognitive processing required in our task, implying notably to

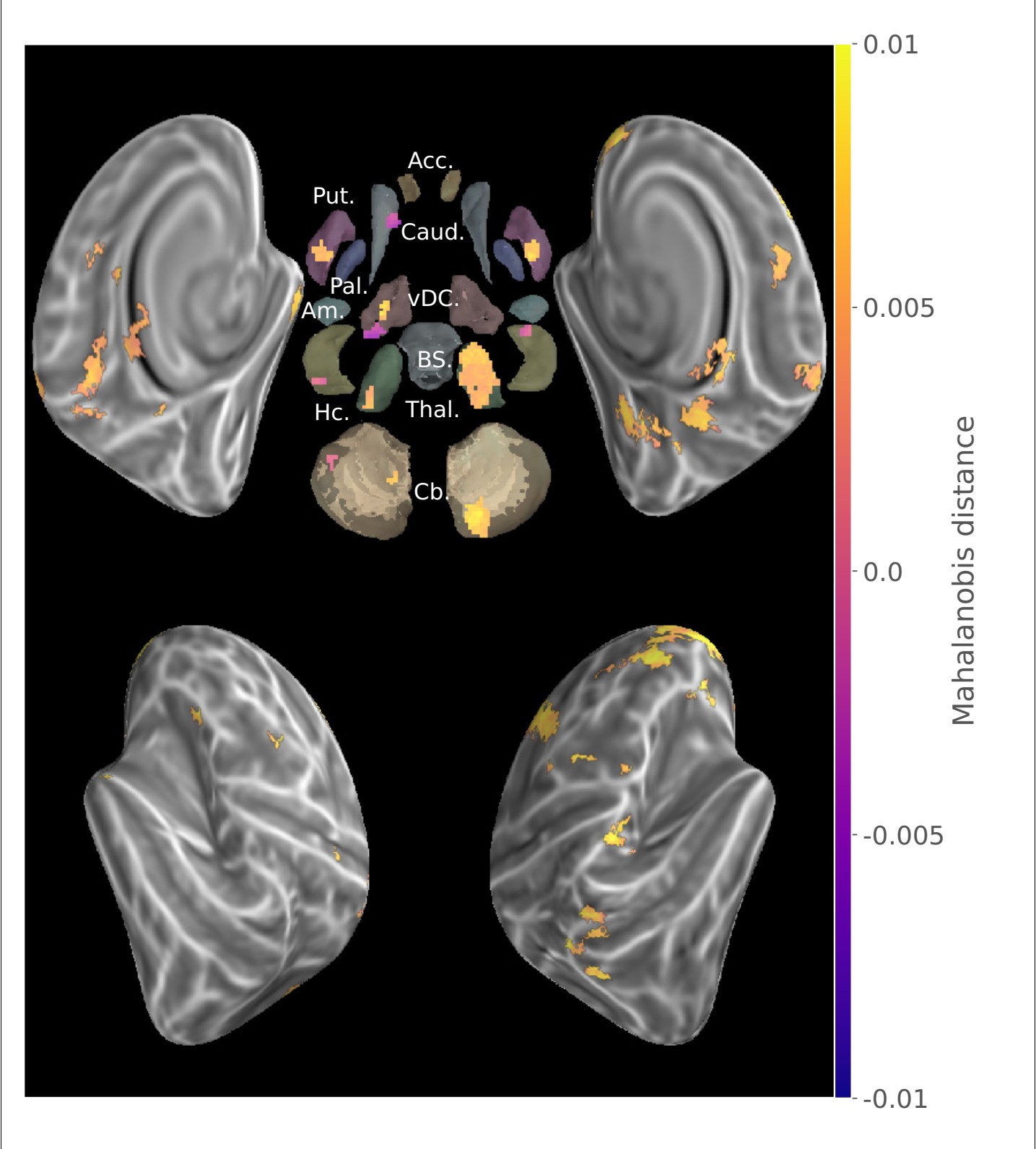

**Figure 3.** Conjunction of group searchlight contrast (paired t-test) between consolidated and new sequences discriminability maps and separate group discriminability maps for new and consolidated sequences (z-score thresholded at p < .05 TFCE-cluster-corrected) showing a reorganization of the distributed memory trace between these two stages. Acc.:Accumbens; Put.:Putamen; Caud.:Caudate; Pal.:Pallidum; vDC:ventral diencephalon; Am.: Amygdala; Hc.:Hippocampus; Thal.:Thalamus; Cb.:Cerebellum; BS:brain-stem.
DOI: https://doi.org/10.7554/eLife.39324.010

switch between sequences, to maintain them in working memory and to inhibit competing ones, could have magnified this frontal associative representation in our study.

In sum, the regions found to carry sequence information regardless of the learning phase in the present study show some overlap with the network known to be implicated in MSL, such as primary and secondary motor cortices, as typically revealed in activation-based studies (*Doyon et al., 2009b*; *Hardwick et al., 2013*; *Dayan and Cohen, 2011*). However, we also found significant representations in the occipital, temporal and insular cortices. This discrepancy can be attributable to the shift from an activation-based inference to one based on the presence of sequential information in activity patterns, but also by the recruitment of additional regions for the processing of this information in stimuli and its maintenance in working memory required by the task.

## Cortico-subcortical representational reorganization underlying memory consolidation following MSL

By contrasting the maps of multivariate distances for consolidated and newly acquired sequences, we identified the set of regions that reveal greater versus lower discriminability of sequential representations in the early stages of the MSL consolidation (*Figure 3*).

At the cortical level, we found that the contralateral premotor and bilateral parietal regions showed a stronger representation for consolidated sequences. This pattern suggests that the tuning of these neural populations to coordinated movements is consolidated early after learning (*Makino et al., 2017*; *Yokoi et al., 2018*; *Pilgramm et al., 2016*), as was previously observed when contrasting a sequence that underwent a longer training with new ones (*Wiestler and Diedrichsen, 2013*). Importantly, no significant changes in representational magnitude were found in the contralateral primary somatosensory cortex after consolidation. This is in line with the fact that M1 representational geometry has been shown to be strongly shaped by ecological finger co-activations (*Ejaz et al., 2015*), and to be resistant to extensive training of a sequence built on a new co-activation structure (*Beukema et al., 2018*). This negative result complements that of *Figure 2*, supporting that primary representation uncovered there might not reflect sequential features but the interaction of pre-existing somatotopic organization and single finger position in each sequences. While the role of the motor cortex in MSL is undeniable, its plasticity in consolidation is still debated (*Omrani et al., 2017*). In fact, recent results revealed that after a M1 insult or even rapidly after M1 inactivation, a trained motor skill can still be expressed (*Kawai et al., 2015*; *Bollu et al., 2018*) arguing for its complementary, redundant and partially independent representation in subcortical regions.

Interestingly, significant differences at the subcortical level were found in bilateral putamen and more specifically in their sensorimotor regions. This is consistent with findings from activation studies that reported increased functional activity after consolidation in this structure (*Debas et al., 2010*; *Albouy et al., 2013b*; *Debas et al., 2014*; *Fogel et al., 2017*; *Vahdat et al., 2017*). As these regions of the basal ganglia are part of anatomical loops with afferents in primary and secondary motor cortices as well as efferents to the cortex via thalamic relays, these can show some degree of somatotopic organization (*Nambu, 2011*) that could contribute to the sequence discriminability, similarly to what is observed in the cortex (*Yokoi et al., 2018*). Significant representational differences were also found in the bilateral thalami, and could reflect the relay of information between the cortex and cerebellum, striatum or spinal regions (*Haber and Calzavara, 2009*; *Doyon et al., 2009a*). Finally, distinct representation levels were detected in the cerebellum, including ipsilateral Lobule IX, shown to correlate with sequential skill performance (*Orban et al., 2010*; *Tomassini et al., 2011*). However, no significant difference was observed in Lobule V of the cerebellum that is known to carry finger somatotopic representations (*Wiestler et al., 2011*) and to show global activation during practice (*Doyon et al., 2002*).

Concurrently with the positive representational differences in the above-mentioned areas, we found only a few disparate regions that showed lower sequence discrimination, namely the caudate nuclei, subthalamic nuclei and cerebellar Crus II ipsilaterally as well as bilateral hippocampi. Hippocampal activation in early learning has formerly been hypothesized to support the temporary storage of new explicitly acquired motor sequence knowledge and to contribute to the reactivations of the distributed network during offline periods and sleep in particular. Yet such contribution of the hippocampus has been shown to be progressively disengaging afterward (*Albouy et al., 2013b*), and thus our results are consistent with the idea of the hippocampus playing a transient supportive role in

early MSL, notably in encoding sequential information (*Davachi and DuBrow, 2015*). Our findings of a differential implication of dorsomedial and dorsolateral striatum in sequence representation during learning and expression of a consolidated skill bring specificity to the changes in activity in these regions in the course of MSL described by earlier studies (*Lehéricy et al., 2005*; *Jankowski et al., 2009*; *François-Brosseau et al., 2009*; *Kupferschmidt et al., 2017*; *Corbit et al., 2017*; *Fogel et al., 2017*; *Reithler et al., 2010*). Indeed, our results uncover that this shift in activity is consistent with animal studies showing a reorganization at the neuronal level in these regions (*Costa et al., 2004*; *Yin et al., 2009*; *Miyachi et al., 2002*).

While our results show regional magnitude of motor sequential information differs between consolidated and newly acquired memory traces, the present study was not designed to investigate the information content of hippocampal, striatal or cerebellar sequence representations. These were previously assessed at the cortical level for finger sequences (*Wiestler et al., 2014*; *Kornysheva and Diedrichsen, 2014*) as well as for larger forearm movements (*Haar et al., 2017*). However, the hypothesized extrinsic and intrinsic skill encoding in the respective hippocampal and striatal systems (*Albouy et al., 2013a*) remains to be assessed with a dedicated experimental design similar to that used by *Wiestler et al. (2014)* to investigate such representations at the cortical level. Moreover, recent research (*Burman, 2018b*; *Burman, 2018a*) pointed out that the hippocampus show significant connectivity with effector-specific regions of the primary motor cortices, without implication of sequential learning, which could reflect its involvement in spatial planning.

Importantly, our study investigated the differentiation of neural substrates of sequence representation after limited training and following sleep-dependent consolidation. This is in contrast to previous investigations that studied sequences trained intensively for multiple days (*Nambu et al., 2015*) and compared their discriminability to that of newly acquired ones (*Wiestler and Diedrichsen, 2013*). Therefore, in our study, the engagement of these representations for expressing the sequential skill may further evolve, strengthen or decline locally with either additional training or offline memory reprocessing supported in part by sleep.

## Methodological considerations

To limit the level of difficulty and duration of the task, only four sequences were performed by participants: two consolidated and two newly acquired. This low number of sequences per condition could be a factor limiting the power of our analysis, as only a single multivariate distance is assessed for each of these conditions. As such, the current experimental design prevented a deeper investigation of the level of the representations in the hierarchical encoding of complex sequential motor skills, from single movement position and chunks to the whole sequence (*Yokoi et al., 2018*). Moreover, initial training sessions of the consolidated sequences were each comprised of a single sequence performed in blocks longer than in the present task, designed for multivariate investigation. The current task, by requiring additional cognitive resources (such as instruction processing, retention in working memory, switching and inhibition of other sequences), could have triggered some novel learning for the consolidated sequences. This seems unlikely however, as this was not reflected in performance changes throughout the task. The switching component could partly explain the pattern of results found here, as shifting between overlapping sets of motor commands has been shown to further implicate the dorsal striatum in collaboration with the prefrontal cortex (*Monchi et al., 2006*). Similarly, due to the main goal of the study, which was to investigate consolidation and reconsolidation of sequential motor learning, the trained sequences were not learned on the same day, potentially inducing a modulating effect on the brain patterns that were measured. However, as reconsolidation is thought to rely on mechanisms similar to consolidation, we believe that both sequences were supported by local networks in the same regions that evoked differentiated patterns of activity rather than by large-scale reconsolidation-induced activity changes.

Another potential limitation relates to the fact that the present representational analysis disregarded the behavioral performance. In particular, it can be noted from *Figure 1* that the subjects' performance of the new sequences evolved as they practiced them. Thus, as we assessed differences in patterns evoked by their execution across the whole session, it is possible that representations emerging when the performance plateaued were missed by our analysis. This problem is inherent to any investigation of learning mechanisms and should be addressed in future studies measuring the dynamic of pattern difference changes within the training session, which would require both a sensitive metric and signal of the highest quality. Regarding influence of performance speed, the chained

non-linear relations between behavior, neural activity and BOLD signal were recently established to have limited influence on the representational geometry extracted from Mahalanobis cross-validated distance in the primary cortex, sampled across a wide range of speed of repeated finger-presses and visual stimulation (*Arbuckle et al., 2019*). Therefore, despite behavioral variability and potential ongoing evolution of the memory trace, we assumed that the previously encoded motor sequence engrams were nevertheless retrieved during this task as supported by the significant differences in activity pattern discriminability and the persistent behavioral advantage observed for the consolidated sequences.

Finally, our results also show that it is possible to investigate learning-related representational changes in a shorter time-frame and with less extended training than what was investigated before (*Wiestler and Diedrichsen, 2013*; *Nambu et al., 2015*), including in subcortical regions where neuronal organization differs from that of the cortex. The use of a novel multivariate distance could have contributed to obtain these results by achieving increased sensitivity and statistical robustness (*Walther et al., 2016*).

## Conclusion

Our study shows that the consolidation of sequential motor knowledge is supported by the reorganization of newly acquired representations across a distributed group of cerebral regions. We uncover that following learning, local activity patterns tuned to represent sequential knowledge are greater not only in extended cortical areas, similarly to those shown after longer training (*Wiestler and Diedrichsen, 2013*), but also in dorsolateral striatum, thalamus and cerebellar regions. Conversely, a smaller set of regions showed a lower sequence-specific patterned activation after consolidation, occurring specifically in the dorsomedial striatum that supports cognitive processing during early learning (*Doyon et al., 2018*) as well as in the hippocampus, which carries explicit encoding of motor sequential extrinsic representation (*Albouy et al., 2013b*; *King et al., 2017*) and play a significant role in the offline reprocessing. Despite discrepancies with GLM-based activity changes observed previously, the results of our novel representational approach corroborate their interpretations that the differential plasticity changes in the latter regions subtend MSL consolidation (*Albouy et al., 2015*). Importantly, these results reveal for the first time in humans that such changes are accompanied by the local implementation of distributed neural coding of sequential information. Yet such consolidation-related representational changes need to be further investigated through exploration of the dynamic mechanism mediating this sleep-dependent mnemonic process, which is known to reorganize progressively the cerebral network by repeatedly reactivating the memory trace (*Boutin et al., 2018*; *Fogel et al., 2017*; *Vahdat et al., 2017*).

## Materials and methods

### Participants

Right-handed young ($n = 34, 25 \pm 6.2$ y), healthy individuals (19 females), recruited by advertising on academic and public website, participated in the study. Participants were excluded if they had a history of neurological psychological or psychiatric disorders, scored four and above on the short version of Beck Depression Scale (*Beck et al., 1961*), had a BMI greater than 27, smoked, had an extreme chronotype, were night-workers, had traveled across meridians during the three previous months, or were trained as a musician or professional typist for more than a year. Their sleep quality was subjectively assessed, and individuals with a score to the Pittsburgh Sleep Quality Index questionnaire (*Buysse et al., 1989*) greater or equal to 5, or daytime sleepiness Epworth Sleepiness Scale (*Johns, 1991*) score greater than 9, were excluded.

Participants included in the study were also instructed to abstain from caffeine, alcohol and nicotine, to maintain a regular sleep schedule (bed-time 10PM–1AM, wake-time 7AM–10AM) and avoid taking daytime nap for the duration of the experiment. In a separate screening session, EEG activity was also recorded while participants slept at night in a mock MRI scanner and gradientsounds were played to both screen for potential sleep disorders and test their ability to sleep in the experimental environment; 18 participants were excluded for not meeting the criterion of a minimum of 20 min in NREM2 sleep. After this last inclusion step, their sleep schedule was assessed by analyzing the data obtained from an actigraph (Actiwatch 2, Philips Respironics, Andover, MA, USA) worn on the wrist

of the non-dominant hand for the week preceding as well as during the three days of the experiment, hence certifying that all participants complied to the instructions.

Among the 34 participants, one did not show within-session improvement on the task, two didn't sleep on the first experimental night, three were withdrawn for technical problems, one did not show up for thefirst experimental session, and one presented a novel MRI contraindication. Thus, among the 26 participants that completed the research project, a group of 18 (14 females, 25 ± 6.2 y) which, by design, followed the appropriate behavioral intervention for the present study, were retained for our analysis.

All participants provided written informed consent and received financial compensation for their participation. This study protocol was approved by the Research Ethics Board of the 'Comitè mixte d'èthique de la recherche - Regroupement en Neuroimagerie du Quèbec' (CMER-RNQ).

## Procedures and tasks

The present study was conducted over three consecutive evenings and is part of an experiment that aimed to investigate the neural substrates mediating the consolidation and reconsolidation of motor sequence memories during wakefulness and sleep that will be reported separately. On each day, participants performed the experimental tasks while their brain activity was recorded using MRI. Their non-dominant hand (left) was placed on an ergonomic MRI-compatible response pad equipped with four keys corresponding to each of the fingers excluding the thumb.

On the first day (D1), participants were trained to perform repeatedly a five-element sequence (TSeq1: 1-4-2-3-1 where one indicates the little finger and four the index finger). The motor sequence was performed in blocks separated by rest periods to avoid fatigue. Apart from a green or a red cross displayed in the center of the screen, respectively instructing the participants to execute the sequence or to rest, there were no other visual stimuli presented during the task. Participants were instructed to execute the sequence repeatedly, and as fast and accurately as possible, as long as the cross was green. They were then instructed to rest for the period of 25 s as indicated by the red cross. No feedback was provided to the participant regarding their performance. Prior to the practice, they were given the instruction that if they realized that they made an error, they should not try to correct it but instead restart practicing from the beginning of the sequence. During each of the 14 practice blocks, participants performed repeatedly 12 motor sequences (i.e. 60 key-presses per block). In case participants made a mistake during sequence production, they were instructed to stop their performance and to immediately start practicing again from the beginning of the sequence until the end of the block. After completion of the training phase, participants were then administered a short retention test about 15 min later, which consisted of a single block comprising 12 repetitions of the sequence. Then the participants were scanned with concurrent EEG and fMRI for approximately two hours while instructed to sleep.

On the second day (D2), participants were first evaluated on the TSeq1 (one block retest) to test their level of consolidation of the motor sequence, and were then trained on a new sequence (TSeq2: 1-3-2-4-1), which was again performed for 14 blocks of 12 sequences each, similarly to TSeq1 training on D1. Again, they were then scanned during sleep while EEG recordings were simultaneously acquired.

Finally, on the third day (D3), participants first performed TSeq1 for seven blocks followed by seven blocks of TSeq2, each block including 12 repetitions of the sequence or 60 key-presses. Following this last testing session, participants were then asked to complete an experimental task (here called MVPA task) specifically designed for the current study, similar to a previous study that investigated sequence representation by means of multivariate classification (*Wiestler and Diedrichsen, 2013*). Specifically, participants performed short practice blocks of four different sequences, including TSeq1 and TSeq2 that were then consolidated, as well as two new finger sequences (NewSeq1: 1-2-4-3-1, NewSeq2: 4-1-3-2-4). In contrast to *Wiestler and Diedrichsen (2013)*, however, all four sequences used only four fingers of the left hand, excluding the thumb. Also, as for the initial training, sequences were instead repeated uninterruptedly and without feedback, in order to probe the processes underlying the forthcoming automatization of the skill.

During this task, each block was composed of an instruction period of 4 s during which the sequence to be performed was visually displayed as a series of five numbers (e.g. 1-4-2-3-1) that could easily be remembered by the participant. The latter was then followed by an execution phase triggered by the removal of the instruction stimuli and the appearance of a green cross on the

screen. In each block, participants performed five times the same sequence (or a maximum of 25 key-presses) without any feedback, before being instructed to stop and rest when the red cross was displayed.

The four sequences were assigned to blocks such as to include all possible successive pairs of the sequences using De-Bruijn cycles (*Aguirre et al., 2011*). This order prevents the systematic leakage of BOLD activity patterns between blocks in this rapid design as across blocks each sequence is preceded and followed by any sequence the same number of times within each run. As a two-length De-Bruijn cycle of the four sequences has to include each sequence four times, this yielded a total of 16 blocks. In our study, two different De-Bruijn cycles were each repeated twice in two separate scanning runs separated by approximately 5 min of rest, hence resulting in a total of 64 blocks (4 partitions of 16 practice blocks, for a total of 16 blocks per sequence). The blocks were synchronized to begin at a fixed time during the TR of the fMRI acquisition.

## Behavioral statistics

Using data from the MVPA task, we entered the mean duration per block of correctly performed sequences into a linear mixed-effect model with a sequence learning stage (new/consolidated) by block (1–16) interaction to test for difference in their performance level, as well as the evolution during the task, with sequences and blocks as random effects and participants as the grouping factor. The same model was run with the number of correct sequences as the outcome variable. Two other models were also used on subsets of data to test separately if there was any significant difference in performance (speed and accuracy) between the two consolidated sequences and between the two new sequences. Linear mixed-effect models were estimated using python package statsmodels (*Seabold and Perktold, 2010*), and outputs are reported in supplementary materials.

## MRI data acquisition

MRI data were acquired on a Siemens TIM Trio 3T scanner with two different setups. The first used a 32-channel coil to acquire high-resolution anatomical T1 weighted sagittal images using a Multi-Echo MPRAGE sequence (MEMPRAGE; voxel size = 1 mm isometric; TR = 2530 ms; TE = 1.64,3.6,5.36,7.22 ms; FA = 7; GRAPPA = 2; FoV= $256 \times 256 \times 176\,mm$) with the different echoes combined using a Root-Mean-Square.

Functional data were acquired with a 12-channel coil, which allowed to fit an EEG cap to monitor sleep after training, and using an echo-planar imaging (EPI) sequence providing complete cortical and cerebellum coverage (40 axial slices, acquire in ascending order, TR = 2160 ms; FoV = $220 \times 220 \times 132$ mm, voxel size = $3.44 \times 3.44 \times 3.3$ mm, TE = 30 ms, FA = 90, GRAPPA = 2). Following task fMRI data acquisition, four volumes were acquired using the same EPI sequence but with reversed phase encoding to enable retrospective correction of distortions induced by B0 field inhomogeneity.

## MRI data preprocessing

High-resolution anatomical T1 weighted images were preprocessed with Freesurfer (*Dale et al., 1999*; *Fischl et al., 1999*; *Fischl et al., 2008*) to segment subcortical regions, reconstruct cortical surfaces and provide inter-individual alignment of cortical folding patterns. Pial and gry/white matter interface surfaces were downsampled to match the 32 k sampling of Human Connectome Project (HCP) (*Glasser et al., 2013*). HCP subcortical atlas coordinates were warped onto individual T1 data using non-linear registration with the Ants software (*Avants et al., 2008*; *Klein et al., 2009*).

A custom pipeline was then used to preprocess fMRI data prior to analysis and relied on an integrated method (*Pinsard et al., 2018*), which combines slice-wise motion estimation and intensity correction followed by the extraction of BOLD timecourses in cortical and subcortical gray matter. This interpolation concurrently removed B0 inhomogeneity induced EPI distortion estimated by the FSL Topup tool using the fMRI data with reversed phase encoding (*Andersson et al., 2003*) acquired after the task. BOLD signal was further processed by detecting whole-brain intensity changes that corresponded to large motion, and each continuous period without such detected event was then separately detrended to remove linear signal drifts. To do so, the whole-brain average absolute value of the BOLD signal derivative was computed and used as a measure to detect widespread signal changes due to large motion. We proceeded by recursively detecting the time

corresponding to the largest global signal change to separate the run data, then detected the second largest change, etc. Once the run data was chunked into periods of stable position constrained to be at least 32 s long, we performed detrending in each chunk separately and concatenated the chunk back together. The rationale behind this procedure is that large movements change the pattern of voxelwise signal trend, and detrending or regression of movement parameters across the whole run can thus induce spurious temporal auto-correlation in the presence of baseline shift. This step was implemented using Python and PyMVPA.

Importantly, the fMRI data preprocessing did not include smoothing, even though the interpolation inherent to any motion correction was based on averaging of values of neighboring voxels. This approach was intended to minimize the blurring of data in order to preserve fine-grained patterns of activity.

## Multivariate pattern analysis

### Samples

Each block was modeled by two boxcars, corresponding to the instruction and execution phases respectively, convolved with the single-gamma Hemodynamic Response Function. Least-square separate (LS-S) regression of each event, which have been shown to provide improved activation patterns estimates for MVPA (*Mumford et al., 2012*), yielded instruction and execution phases beta maps for each block that were further used as MVPA samples.

### Cross-validated multivariate distance

Similarly to *Wiestler and Diedrichsen (2013)* and *Nambu et al. (2015)*, we aimed to uncover activity patterns that represented the different sequences performed by the participants. However, instead of calculating cross-validated classification accuracies, we opted for a representational approach by computing the multivariate distance between activity patterns evoked by the execution of sequences, in order to avoid ceiling effect and baseline drift sensitivity (*Walther et al., 2016*). In the current study, we computed the cross-validated Mahalanobis distance (*Nili et al., 2014*; *Walther et al., 2016*; *Diedrichsen et al., 2016*), which is an unbiased metric that uses multivariate normalization by estimating the covariance from the GLM fitting residuals and regularizing it through Ledoit-Wolf optimal shrinkage (*Ledoit and Wolf, 2004*). Cross-validation was performed across the four balanced data partitions (4 repetitions of each sequence) that correspond to separate DeBruijn cycles. This distance, which measures discriminability of conditions, was estimated separately for pairs of sequences that were in a similar acquisition stage, that is, for the newly acquired and consolidated sequences.

### Searchlight analysis

Searchlight (*Kriegeskorte et al., 2006*) is an exploratory technique that applies MVPA repeatedly on small spatial neighborhoods covering the whole brain while avoiding high-dimensional limitations of multivariate algorithms. Searchlight was configured to select for each gray-matter coordinate their 64 closest neighbors as the subset of features for representational distance estimation. The neighborhood was limited to coordinates in the same structure (hemisphere or region of interest), and proximity was determined using respectively Euclidian (volume-based) and geodesic (surface-based) distance for subcortical and cortical coordinates. The extent of the searchlight was thus kept to such a local range to limit the inflation of false positive or negative results (*Etzel et al., 2004*; *Etzel et al., 2013*).

### Statistical testing

To assess statistical significance of multivariate distance and contrasts, group-level Monte-Carlo non-parametric statistical testing using 10000 permutations was conducted on searchlight cross-validated mahalanobis maps with Threshold-Free-Cluster-Enhancement (TFCE) correction (*Smith and Nichols, 2009*). The statistical significance level was set at $p < .05$ (with confidence interval $\pm.0044$ for 10000 permutations) with a minimum cluster size of 10 features. For the contrast between consolidated and new sequences we applied a minimum-statistic conjunction test (*Nichols et al., 2005*) (equivalent to masking positive and negative differences with the respective simple effect) to assess that the differences found were supported by statistically significant above-zero Mahalanobis distances.

Our BOLD sampling of sparse gray-matter coordinates induces large differences in the sizes of the structures being investigated, and thus conventional cluster corrections based on whole-brain cluster-size distribution would have biased the results. Therefore, TFCE enabled a locally adaptive statistics and cluster-size correction that particularly fitted this sampling.

The MVPA analysis was done using the PyMVPA software (*Hanke et al., 2009*) package with additional development of custom samples extraction, cross-validation scheme, efficient searchlight and multivariate measure computation, optimally adapted to the study design and the anatomy-constrained data sampling.

## Acknowledgments

We thank J Diedrichsen for methodological advice on our multivariate representational analysis.

## Additional information

### Funding

| Funder | Grant reference number | Author |
| --- | --- | --- |
| Canadian Institutes of Health Research | MOP 97830 | Basile Pinsard<br>Arnaud Boutin<br>Ella Gabitov<br>Julien Doyon |
| Ministère de l'Education Nationale, de l'Enseignement Superieur et de la Recherche | PhD scholarship | Basile Pinsard |
| Sorbonne Université | PhD study abroad grant | Basile Pinsard |

The funders had no role in study design, data collection and interpretation, or the decision to submit the work for publication.

### Author contributions

Basile Pinsard, Conceptualization, Software, Formal analysis, Investigation, Visualization, Methodology, Writing—original draft, Writing—review and editing; Arnaud Boutin, Ella Gabitov, Conceptualization, Investigation, Writing—review and editing; Ovidiu Lungu, Conceptualization, Project administration, Writing—review and editing; Habib Benali, Supervision, Methodology; Julien Doyon, Conceptualization, Supervision, Project administration, Writing—review and editing

### Author ORCIDs

Basile Pinsard http://orcid.org/0000-0002-4391-3075
Arnaud Boutin http://orcid.org/0000-0002-5696-2626
Julien Doyon https://orcid.org/0000-0002-3788-4271

### Ethics

Human subjects: All participants provided written informed consent and received financial compensation for their participation. This study protocol was approved by the Research Ethics Board of the "Comité mixte d'éthique de la recherche - Regroupement en Neuroimagerie du Québec" (CMER-RNQ 13-14-011).

### Decision letter and Author response

Decision letter https://doi.org/10.7554/eLife.39324.020
Author response https://doi.org/10.7554/eLife.39324.021

## Additional files

### Supplementary files
• Supplementary file 1. Test for differences in speed (mean duration to perform a correct sequence) per block between trained and new sequences.
DOI: https://doi.org/10.7554/eLife.39324.012

• Supplementary file 2. Test for differences in accuracy (number of correct sequences over the five repetitions in a block) between trained and new sequences.
DOI: https://doi.org/10.7554/eLife.39324.013

• Supplementary file 3. Test for differences in speed and accuracy between the new sequences.
DOI: https://doi.org/10.7554/eLife.39324.014

• Supplementary file 4. Test for differences in speed and accuracy between the consolidated sequences.
DOI: https://doi.org/10.7554/eLife.39324.015

• Source code 1. MNI coordinates of main-effect clusters center-of-mass.
DOI: https://doi.org/10.7554/eLife.39324.016

• Source code 2. MNI coordinates of contrast clusters center-of-mass.
DOI: https://doi.org/10.7554/eLife.39324.017

• Transparent reporting form
DOI: https://doi.org/10.7554/eLife.39324.018

### Data availability
Behavioral data analyzed and presented in the article as well as statistical maps of brain representational measure have been deposited on the Open Science Framework with the DOI 10.17605/OSF.IO/EPJ2V

The following dataset was generated:

| Author(s) | Year | Dataset title | Dataset URL | Database and Identifier |
|---|---|---|---|---|
| Pinsard B | 2018 | Consolidation alters motor sequence-specific distributed representations | https://osf.io/epj2v/?view_only=cca0307cb0d84787b4-b203451e96c0ed | Open Science Framework, 10.17605/OSF.IO/EPJ2V |

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
