## [Decision Letter]

Thank you for submitting your article "Consolidation alters motor sequence-specific distributed representations" for consideration by *eLife*. Your article has been reviewed by three peer reviewers, including Timothy Verstynen as the Reviewing Editor and Reviewer #1, and the evaluation has been overseen by Rich Ivry as the Senior Editor. The following individuals involved in review of your submission have agreed to reveal their identity: Atsushi Yokoi (Reviewer #2).

The reviewers have discussed the reviews with one another and the Reviewing Editor has drafted this decision to help you prepare a revised submission.

Summary:

The present work examines motor skill learning by using multi-voxel pattern analysis to compare patterns of the brain activity elicited by two well-trained sequences that were consolidated by at least one overnight sleep and two novel sequences. The key findings were that while there was some overlap between brain regions in which the two learning stages of sequences were represented (e.g.., bilateral premotor and parietal regions), there were regions where the sequences of either one of the stages were more strongly represented, especially around the sub-cortical structures. These sub-cortical structures included, for example, some parts of bilateral hippocampi, ipsilateral cerebellar cortex, and ipsilateral caudate, where the new sequences were more strongly represented. On the other hand, in the areas including bilateral sensorimotor putamen and thalamus the consolidated sequences were more strongly represented, suggesting that neural substrate storing sequence information drastically changes over the early consolidation process.

Essential revisions:

All three reviewers highlighted major concerns that fall under seven general themes.

1) Performance or representational differences?

The behavioral results clearly show that the performance (both speed and accuracy) is different between the trained and untrained sequences. Thus, the differences in multivariate patterns could be driven by differences in performance rather than differences in encoding. There is (at least) one way to look at this issue: The behavioral performance shown in Figure 1 reveals that performance for the untrained sequences steadily improves across runs and appears to reach an asymptote half-way through the experiment. If the difference in multivariate patterns is truly related to long-term consolidation, as opposed to being the consequent result of changes in performance, then comparing the last 8 blocks of the trained vs. untrained sequences should replicate the key results (i.e., Figure 3). In contrast, comparing the last 8 blocks of the untrained sequences to the first 8 blocks of the of the untrained sequences should not reveal a similar set of clusters. However, if this split-half comparison on the untrained sequences produces qualitatively similar maps as the trained vs. untrained comparison, then it would strongly suggest that the differences in multivariate distances is driven largely by performance effects.

2) Elaboration of representational distances.

The true power of the RSA approach is that you can directly measure the distinctiveness of representations across conditions. Yet it is used mainly here as an alternative to traditional decoding methods. It would be nice for the authors to show the representational distances across sequences in key regions (e.g., cortical motor clusters, striatum, cerebellum). This would give the reader a sense of how training may be altering sequence-related representations.

3) Distinguishing between signal amplitude and representational discriminability.

The map of clusters that discriminate between any of the four sequences (Figure 2), reveals a pretty standard sensorimotor network. Is RSA discriminability really driven by regions with significant task-related BOLD responses as estimated from traditional univariate GLM maps (in contrast to true differences in the covariance pattern of local voxels)? How well does the discriminability of the searchlight correlate with local task-related activity maps from univariate GLM? If they are correlated, how can you distinguish between searchlight results just being the result of local signal-to-noise differences from results driven by true differences in encoding?

Related to this was concern that, even after accounting for the potential first-finger effect, there remains potential differences in overall activity levels across the new and old sequences. Can this be addressed, especially given the differences in behavioural performance between new and consolidated sequences. How much were the overall activities different across the blocks and sequences?). Note that the pair-wise dissimilarities between fingers changed quite a bit in different activation levels (= tapping frequencies) (see Figure 4A), indicating that the direct comparison between thumb/index distance in one tapping frequency and index/little one in another frequency would not be meaningful. The authors could try, for instance, using multivariate pattern angle, which is less affected by pattern scaling (Walther et al., 2016), or assessing the XOR (disjunctive union) of the prevalence of cross-validated distances for the consolidated and the new sequences, avoiding the direct comparison between them.

4) Elaboration of learning mechanisms & dynamics.

The authors make an extensive effort in the Introduction and Discussion sections to try to link these results to hippocampus. However, there is not a direct assessment of the role of the hippocampus to the rest of the network. There is only the observation of a cluster in each hippocampus that reliably distinguishes between the trained and untrained sequences, not an analysis that shows this is driving the rest of the changes in encoding in other areas.

5) Issues with behavioral data:

One reviewer noted that you have referenced a recent paper that shows movement rate differences that are performed with a single digit do not appreciably alter classification results. However, there are likely speed accuracy tradeoff differences between the samples, which may bias classification comparisons.

Moreover, there is no evidence here for consolidation. While there is a significant difference between condition for the trained and novel sequences, the trained were reactivated prior to imaging. Might the advantage for trained have come from the warm-up session (which was not given for the novel)? Performance for the novel sequences becomes similar to the TSeqs after block 12, which corresponds to 60 trials of practice for either novel sequence. The performance distinction between conditions may be entirely driven by the warmup period. What did performance look like for the trained sequences at the very start of this reactivation/warmup period?

6) Controlling for first-finger effects.

As you note in the Discussion section, the pattern discriminability between the sequences starting with different fingers might reflect a "first-finger effect", where the discriminability of two sequences is almost solely driven by which finger was moved the first, not by the sequential order per se. This also applies to the pattern dissimilarity between the two new sequences (1-2-4-3-1 and 4-1-3-2-4) which, in contrast to the two consolidated sequences that had the same first finger (1-4-2-3-1, and 1-3-2-4-1), had different first fingers. Without accounting for this potential confound, the comparison made between the "new/new" and the "old/old" dissimilarities is hard to interpret, as it is unclear whether we are comparing between genuine sequence representations, or between sequence representation and individual finger representation.

7) Clarification of methods.

How did participants know that they made an error? Were they explicitly told a key press was incorrect? Or did they have to self-monitor performance during training? Was the same button box apparatus used during scanning as in training? Was the presentation of sequence blocks counterbalanced during scanning? How many functional runs were performed? Is the classification performance different between runs?

Additional detail is needed regarding the correction of motion artefact. In subsection “MRI data preprocessing”, the authors state that BOLD signal was further processed by detecting intensity changes corresponding to "large motion". What additional processing was performed? What was considered large motion? Was motion DOFs included in the GLM? Are there differences in motion between sequences and between sequence conditions? More information is needed on the detrending procedure. Is there evidence that detrending worked similarly between the different time windows between spikes? How were spikes identified? What happened to the spikes? In general, what software was used for the imaging analysis? Just PyMVPA?

The description of the analysis in subsection “Reorganization of the distributed sequence representation after memory consolidation” should be reworded. Is this describing how you analyze the difference in the consolidated and novel sequence discriminability maps? But how is this a conjunction? A conjunction reflects the overlap between 2 contrasts, and in this case what we are looking at is a difference. Related to this, there are different types of conjunctions. Please provide more details, as conjunctions can inflate significance. What software was used and how were the thresholds set for the conjunction?

[Editors' note: further revisions were requested prior to acceptance, as described below.]

Thank you for resubmitting your work entitled "Consolidation alters motor sequence-specific distributed representations" for further consideration at *eLife*. Your revised article has been favorably evaluated by Richard Ivry (Senior Editor), Tim Verstynen (Reviewing Editor), and two reviewers.

The manuscript has been improved but there are some remaining issues that need to be addressed. While we like to minimize back and forth in the review process, we believe the first two issues are critical here, especially the first, in establishing the robustness of your effects.

Essential revisions:

1) Concern that the statistical tests are too liberal. The contrasts appear to be global null conjunction tests. There has been significant debate regarding the usage of global null conjunction in fMRI research (see commentary on conjunctions involving papers by Nichols and Friston, 2006). Conjunctions of the global null do not require that activation for both (or however many) individual maps supplied to the conjunction to be individually significant, and as result, small effects can be amplified. Further, most often in the literature conjunctions are performed between multiple differential condition contrasts (e.g., CondA>CondB with CondA>CondC), and not (CondA>CondB with CondA>baseline). The primary concern here is with the contrast involving differences between consolidated and novel sequences. The authors test: (consolidated > new) ∧ (consolidated > 0); and then for new: (new > consolidated) ∧ (new > 0). It is unclear why contrasts relative to baseline are integrated in the contrast. We recognize that there is reason to want to correct for effects that are positive, but a more parsimonious version of this would be to use a liberal mask for positive effects, using the contrast (consolidated > new), where there is no risk of inflating statistics. Instead, by incorporating the baseline contrast in the conjunction the effect is driven by (consolidated > 0), and not the more subtle effects that are of interest (e.g., consolidated > new). It is critical that the authors demonstrate the effects are present without the baseline conjunction test. Again, there is no need for a conjunction. Simple contrasts (consolidated > new, masked by consolidated > 0) and vice versa seem to be a preferable solution and we would like to see this in lieu of the conjunction analysis.

2) In the retest session before the MVPA session, is there an overnight gain in performance (i.e., gain in performance measure between the last trial of previous day and the first trial of the next day) in the two trained sequences? This is the typical behavioural evidence for motor memory consolidation (e.g., Stickgold, 2005), and we think it should be incorporated here.

3) The consolidated sequences were not learned at the same time and not subject to the same consolidation and reconsolidation period. These differences could ultimately bias the classification results for the consolidated sequences. Tseq1 was practiced on Day1, and then tested on Day2. Tseq2 was then immediately practiced after Tseq1. Tseq1 and Tseq2 were tested in the scanner protocol on Day3. There are 2 problems. First, Tseq1 was consolidated over 2 nights of sleep, versus 1 for Tseq2. Because of this, any difference in neural representation could be due to the simple age and consolidation history of the memory. Second, exposure to a new sequence (Tseq2) immediately following the reactivation of Tseq1 can interfere with the reconsolidation of Tseq1. Evidence from recent studies of motor reconsolidation have shown that exposure to similar manipulation can significantly alter the memory trace for the reactivated, previously consolidated memory (Tseq1). Essentially what is left is a classification between a sequence that was consolidated and reconsolidated vs. a sequence that was consolidated. It is unclear why the authors did not train both of the sequences at the same time particularly because they are testing them as such. This limitation should be addressed.

4) Regarding the performance issue (i.e., improvement of new sequences during the MVPA session): Critically, in their response, the authors argued that the inconsistency of Author response image 1 to Figure 3 is due to the lack of statistical power because of using only one cross-validation fold. However, at the same time, they argued that the inconsistency of Author response image 2 (the contrast of distances between the first and the second run) to Figure 3 is the supporting evidence that the result presented in Figure 3 reflects consolidation, despite that these estimates were also computed using only one cross-validation fold. If they say Author response image 1 is unreliable because of using single fold, then what is presented in the Author response image 2 should also be unreliable for the very same reason. Ideally, the new sequences should have practiced well until their performance and underlying representations reached to plateau before going into the scanner, as the initial rapid change in the representation would essentially have little thing to do with consolidation process. It is still true that the distance estimate computed through the 4 partitions would be significantly biased if there is some inconsistency in pattern difference in any of the partitions due to the ongoing performance improvement. Therefore, caution is required to conclude that the difference highlighted in Figure 3 can be attributed specifically to the *consolidation*. The authors should discuss the potential impact of the sub-optimal estimate of the plateau representation for the new sequences.

---

## [Author Response]

Essential revisions:All three reviewers highlighted major concerns that fall under seven general themes.1) Performance or representational differences?The behavioral results clearly show that the performance (both speed and accuracy) is different between the trained and untrained sequences. Thus, the differences in multivariate patterns could be driven by differences in performance rather than differences in encoding. There is (at least) one way to look at this issue: The behavioral performance shown in Figure 1 reveals that performance for the untrained sequences steadily improves across runs and appears to reach an asymptote half-way through the experiment. If the difference in multivariate patterns is truly related to long-term consolidation, as opposed to being the consequent result of changes in performance, then comparing the last 8 blocks of the trained vs. untrained sequences should replicate the key results (i.e., Figure 3). In contrast, comparing the last 8 blocks of the untrained sequences to the first 8 blocks of the of the untrained sequences should not reveal a similar set of clusters. However, if this split-half comparison on the untrained sequences produces qualitatively similar maps as the trained vs. untrained comparison, then it would strongly suggest that the differences in multivariate distances is driven largely by performance effects.

The reviewers are right that behavioral differences between trained and untrained task are an inherent limitation common to most design investigating learning, and particularly the fact that performances of an untrained task tend to improve while being measured.

To further support our results, however, we conducted the suggested analyses:

The contrast between trained and untrained sequence distances in the last run (i.e. when performance for both sequences reached an asymptote) resulted in significant differences illustrated in Author response image 1 which only partly replicates the results of Figure 3. Notably only contralateral putamen representation is found significantly different, while no significant difference is found in the hippocampus. Among the potential factors that could have influenced these results, is the important reduction in statistical power induced by going from a 6-folds to a 1-fold measure.

**Author response image 1. respfig1:** Contrast between within-trained and within-consolidated sequence mahalanobis distances in the second acquisition run (p<.05 Monte-Carlo TFCE corrected).

Contrasting untrained sequence across runs, provided results (see Author response image 2) distinct from that displayed in Figure 3, in that pattern difference was weaker in the second run as compared to the first in a group of regions including the anterior parts of the caudate bilaterally as well as the contralateral putamen and dorsal prefrontal cortex. While being out of the scope of the present manuscript and conjectural, these results could reflect the quickly decreasing implication of the anterior cortico-striatal attentional processes in the initial “fast-learning” phase of MSL.

**Author response image 2. respfig2:** Contrast (run2-run1) between new sequences Mahalanobis distances (p<.05 Monte-Carlo TFCE corrected).

Therefore, these analyses support the fact that our results reflect stable pattern differences across the whole session, despite the ongoing acquisition of the untrained sequences and inherent changes in behavioral performance.

Moreover, the use of cross-validated Mahalanobis distance between patterns evoked by the production of sequence at the same stage of learning does have advantages over classification measures regarding potential behavioral bias. More precisely, it measures how these pattern differences within splits are stable across splits of the data. Thus, pattern differences have to be consistently present within each split in which behavioral measures of the untrained sequences do not differ. It also has to be noted that among the 6 folds of the cross-validation, 4 are across the 2 runs, and thus assess which patterns differences are stable across runs rather independently of the abovementioned behavioral changes. Taken together, we thus think that the stable pattern difference observed reflects true encoding of persistent sequential information.

2) Elaboration of representational distances.The true power of the RSA approach is that you can directly measure the distinctiveness of representations across conditions. Yet it is used mainly here as an alternative to traditional decoding methods. It would be nice for the authors to show the representational distances across sequences in key regions (e.g., cortical motor clusters, striatum, cerebellum). This would give the reader a sense of how training may be altering sequence-related representations.

We agree with the reviewer that the present study uses RSA as a continuous alternative to classification-based decoding. As a matter of fact, our study was initially designed for the latter, but RSA was then used to avoid the bias and limitation of classification. As such, we only included two sequences per condition (consolidated/new) resulting in one distance per condition, and this prevents us from investigating the geometry of representations.

If we understand the reviewer’s comment, the addition of ROIs distance matrix in results would benefit the present study. However, in this 4x4 matrix, only the two within-condition measures which are presented on brain surface is interpretable. We replaced the statistical measure in the maps by the group-average cross-validated Mahalanobis distance. We hope that this modification provides a better sense of the changes in sequence representations.

3) Distinguishing between signal amplitude and representational discriminability.The map of clusters that discriminate between any of the four sequences (Figure 2), reveals a pretty standard sensorimotor network. Is RSA discriminability really driven by regions with significant task-related BOLD responses as estimated from traditional univariate GLM maps (in contrast to true differences in the covariance pattern of local voxels)? How well does the discriminability of the searchlight correlate with local task-related activity maps from univariate GLM? If they are correlated, how can you distinguish between searchlight results just being the result of local signal-to-noise differences from results driven by true differences in encoding?

The reviewers are right that the clusters found in Figure 2 are partly consistent with univariate results, and it is expected that representation of information relevant to sequence production are often present in a subset of the regions that are significantly activated by the task. In a sense, any RSA result is a ratio of the strength or variance that enables discriminating the information of interest over the level of noise.

As suggested, we conducted a GLM analysis on the activity evoked during motor sequence production and the resulting map (see Author response image 3) reproduces a well-known motor network, but it does not exactly correlate with the results of Figure 2. Notably, subcortical activity common to both conditions does not result in common level of sequence representation.

The main effect GLM analysis was added in the supplementary material and the following sentence was added to the manuscript (subsection “A common distributed sequence representation irrespective of learning stage”): “To complement this result we extracted the main effect of activation during this task (Figure 1—figure supplement 3), showing that pattern distance do not exactly correspond to the regions significantly activated in mass-univariate GLM analysis.”

When contrasting consolidated and new sequence activity during execution, no result survived FDR correction, and thus, our results do not seem to result from large-scale activity scaling over the extent of each searchlight. It has to be noted that our data were not smoothed during preprocessing, and that this might have prevented the assessment of large-scale activity differences between conditions recruiting similar networks. Nonetheless, this approach allowed us to measure fine-grained pattern differences.

The following sentence was added to the manuscript (subsection “Reorganization of the distributed sequence representation after memory consolidation”): “We also conducted a univariate GLM analysis to contrast the consolidated and new sequences, but no results survived multiple comparison correction. This negative result could be accounted by the absence of smoothing of signals during preprocessing.”

**Author response image 3. respfig3:** Main effect of motor sequence execution during MVPA task. (t-value thresholded at p<.05 FDR-corrected).

Related to this was concern that, even after accounting for the potential first-finger effect, there remains potential differences in overall activity levels across the new and old sequences. Can this be addressed, especially given the differences in behavioural performance between new and consolidated sequences. How much were the overall activities different across the blocks and sequences?). Note that the pair-wise dissimilarities between fingers changed quite a bit in different activation levels (= tapping frequencies) (see Figure 4A), indicating that the direct comparison between thumb/index distance in one tapping frequency and index/little one in another frequency would not be meaningful. The authors could try, for instance, using multivariate pattern angle, which is less affected by pattern scaling (Walther et al., 2016), or assessing the XOR (disjunctive union) of the prevalence of cross-validated distances for the consolidated and the new sequences, avoiding the direct comparison between them.

Following the reviewer’s suggestion, we thresholded the contrast map with the following xor test: ((Consolidated>0)⊕(New>0))∩(Consolidated>New)) which is presented in Author response image 4. Interestingly, the results are similar to that of Figure 3. It thus appears that the changes we reported are less likely to result from the scaling of the “pattern-to-noise” ratio.

While not being certain, we here assume the reviewer is referring to Arbuckle et al., 2018, when mentioning Figure 4A. We agree that their results show a dissimilarity scaling in relation the frequency of single-finger tapping that can alternatively be explained by the number of movements/key presses per trial (Figure 1). While they manipulated frequency in their study, it is probable that the number of movements is scaling the neuronal activity and, indirectly, the BOLD signal, which results in an increased signal-to-noise ratio. In our protocol, the number of key presses was fixed to be the same across all trials. However, their frequency significantly evolved for the untrained sequences, and remained significantly different from the frequency of the trained sequences across the task. Even if this difference does not amount to doubling frequency as in (Arbuckle et al., 2018), it could still influence our results, notably during the first run where changes are more pronounced. Contrary to Arbuckle et al., (2018), trial lengths in the present study were inversely related to the tapping frequency, and thus, patterns of slower untrained sequences average slightly more TR than patterns of trained sequences. Longer trials should therefore benefit from increased signal-to-noise ratio. In sum, there are many co-varying behavioral and statistical factors that can influence RSA measures. The Arbuckle article is a very interesting step in characterizing these.

**Author response image 4. respfig4:** Conjunction of Consolidated>New contrast map and exclusive disjunction of Consolidate>0 and New>0 maps (p<.05 Monte-Carlo TFCE corrected).

4) Elaboration of learning mechanisms & dynamics.The authors make an extensive effort in the Introduction and Discussion sections to try to link these results to hippocampus. However, there is not a direct assessment of the role of the hippocampus to the rest of the network. There is only the observation of a cluster in each hippocampus that reliably distinguishes between the trained and untrained sequences, not an analysis that shows this is driving the rest of the changes in encoding in other areas.

The approach that we adopted is a data-driven mapping technique that explores the localized representations of information using an alternative metric and, as such, does not model interactions between regions. The results revealed the stronger implication of the hippocampus in the initial learning phase, a finding that we then discussed in relation to previous studies with more conventional GLM-based and connectivity measures.

We removed references to our results as a network, as we are not conducting any kind of network analysis with connectivity measures. We hope that this clarification improves the reader's understanding of our approach and answers the reviewers’ concerns.

5) Issues with behavioral data:One reviewer noted that you have referenced a recent paper that shows movement rate differences that are performed with a single digit do not appreciably alter classification results. However, there are likely speed accuracy tradeoff differences between the samples, which may bias classification comparisons.Moreover, there is no evidence here for consolidation. While there is a significant difference between condition for the trained and novel sequences, the trained were reactivated prior to imaging. Might the advantage for trained have come from the warm-up session (which was not given for the novel)? Performance for the novel sequences becomes similar to the TSeqs after block 12, which corresponds to 60 trials of practice for either novel sequence. The performance distinction between conditions may be entirely driven by the warmup period. What did performance look like for the trained sequences at the very start of this reactivation/warmup period?

We agree with the reviewers that our results could be confounded by the retest of trained sequences, which were performed prior to the task of interest in the present work. However, the statistics show that the sequence duration is lower for trained sequences as compared to new sequences throughout the two runs. It is true that, visually in Figure 1, the gap seems to be closed around block #12. However, the remaining difference is consistent with MSL literature that shows small marginal gains during consolidation as compared to the improvement observed during training. While Figure 1 shows the between-subject variance with error-bars, the mixed-effect model approach does take separately into account between-subject variance, and as such, within-subject differences between conditions are assessed (see supplementary materials).

Following the reviewers’ suggestion, we extracted the mean duration of the first 5 sequences (over the 12 sequences) of each block during the retest of trained sequences. This was done in order to make this measure more comparable to the MVPA task, as higher speed is facilitated by longer blocks. We then plotted the sequences (see Author response image 5) and performed statistical comparison of the first block mean duration between trained and new sequences. The analyses revealed that the trained sequences show some warm-up during the first blocks of retest, yet the speed in the first block is significantly greater (β = -.431, SE = 0.090, p < 10^-5^) than the speed of new sequences at the beginning of MVPA task. Therefore, we believe that consolidation of the trained sequences did occur. We added the following sentence to integrate these results in the manuscript (subsection “Behavioral performance”): “To verify that behavioral differences could not be accounted by the preceding retest of trained sequences, we compared the sequence duration of the first 5 sequences of the retest blocks to the duration of the untrained sequences during the present task. The values reported in supplementary material (Figure 1—figure supplement 1), show that the trained sequence are still performed faster than the untrained sequences (β = -.431, SE = 0.090, p < 10^-5^).

**Author response image 5. respfig5:** Sequence duration of Trained sequences in the first blocks for each sequence on the last experimental day (mean and standard error across participants).

6) Controlling for first-finger effects.As you note in the Discussion section, the pattern discriminability between the sequences starting with different fingers might reflect a "first-finger effect", where the discriminability of two sequences is almost solely driven by which finger was moved the first, not by the sequential order per se. This also applies to the pattern dissimilarity between the two new sequences (1-2-4-3-1 and 4-1-3-2-4) which, in contrast to the two consolidated sequences that had the same first finger (1-4-2-3-1, and 1-3-2-4-1), had different first fingers. Without accounting for this potential confound, the comparison made between the "new/new" and the "old/old" dissimilarities is hard to interpret, as it is unclear whether we are comparing between genuine sequence representations, or between sequence representation and individual finger representation.

The reviewers are right that this is a limitation for part of the current results, as discussed in the original manuscript (now subsection “A distributed representation of finger motor sequence”). However, the design of the present task is different of the one that uncovered this effect in that each sequence is executed separately and the execution is uninterrupted. This should theoretically reduce considerably this initiating effect. Moreover, the main results of the present paper are the differences in striatal and hippocampal representations, and these regions are not expected to show strong and reliable somatotopy for individual finger movements. Therefore, we do not expect a pattern associated with the first finger in these regions. While this effect is known to impact the motor cortical regions, it would be expected to bias positively the new sequences discriminability, but no cortical regions show higher discriminability than for consolidated sequences. At most, this effect could have caused some false-negative results in the contrast between conditions. As such, we still believe that our results are of interest for shedding light in our understanding of the neural substrate mediating motor sequence learning.

7) Clarification of methods.How did participants know that they made an error? Were they explicitly told a key press was incorrect? Or did they have to self-monitor performance during training? Was the same button box apparatus used during scanning as in training? Was the presentation of sequence blocks counterbalanced during scanning? How many functional runs were performed? Is the classification performance different between runs?

No feedback was provided to the participant regarding their performance. Prior to the practice period, they were given the instruction that if they realized that they made an error, they should not try to correct it but instead to restart practicing from the beginning of the sequence. All training and testing took place in the scanner with the same apparatus (subsection “Procedures and tasks”). The order of sequence blocks was determined by DeBruijn cycles that allow to counterbalance the temporal successions of conditions across each run (subsection “Procedures and tasks”). Two functional runs were performed and each run contained two repetitions of the DeBruijn counterbalancing and were thus split to get four folds of cross-validation. The “classification performance” or rather pattern discriminability was not contrasted between runs as it was not the scope of the present study, but representational changes could occur for untrained sequences in a subset of regions.

As we’ve now come to realise that the information pertaining to error feedback was not explicitly described in the original manuscript, the following sentences have been added to subsection “Procedures and tasks”: "No feedback was provided to the participant regarding their performance. Prior to the practice period, they were given instructions that if they realized they made an error, they should not try to correct it but instead restart practicing from the beginning of the sequence."

Additional detail is needed regarding the correction of motion artefact. In subsection “MRI data preprocessing”, the authors state that BOLD signal was further processed by detecting intensity changes corresponding to "large motion". What additional processing was performed? What was considered large motion? Was motion DOFs included in the GLM? Are there differences in motion between sequences and between sequence conditions? More information is needed on the detrending procedure. Is there evidence that detrending worked similarly between the different time windows between spikes? How were spikes identified? What happened to the spikes? In general, what software was used for the imaging analysis? Just PyMVPA?

We agree that the manuscript would benefit from a more detailed account of the motion artefact correction.

The motion artifact removal was performed independently from the GLM regression. A detrending approach designed to avoid introducing long-range temporal correlation that are detrimental to cross-validation was adopted. The description of the detrending procedure was extended with the following sentence (subsection “MRI data preprocessing”):

"To do so, the whole-brain average absolute value of the BOLD signal derivative was computed and used as a measure to detect widespread signal changes due to large motion. We proceeded by recursively detecting the time corresponding to the largest global signal change to separate the run data, then detected the second largest change, etc. Once the run data was chunked into periods of stable position constrained to be at least 32 sec. long, we performed detrending in each chunk separately and concatenated the chunk back together. The rationale behind this procedure is that large movements change the pattern of voxelwise signal trend, and detrending or regression of movement parameters across the whole run can thus induce spurious temporal auto-correlation in presence of baseline shift. This step was implemented using Python and PyMVPA."

The description of the analysis in subsection “Reorganization of the distributed sequence representation after memory consolidation” should be reworded. Is this describing how you analyze the difference in the consolidated and novel sequence discriminability maps? But how is this a conjunction? A conjunction reflects the overlap between 2 contrasts, and in this case what we are looking at is a difference. Related to this, there are different types of conjunctions. Please provide more details, as conjunctions can inflate significance. What software was used and how were the thresholds set for the conjunction?

We agree with the reviewers that this sentence would benefit from a clarification. The contrast map was thresholded by the union of two conjunctions

((Consolidate>New)∩(Consolidated>0))∪((New>Consolidated)∩(New>0)). Each of these subtests was performed on the p-value resulting from non-parametric permutation testing with p-values thresholded at.05. As we are using logical conjunctions or minimum statistics, no inflation is expected. The permutation testing was implemented in Python.

We rephrased it as such (subsection “Reorganization of the distributed sequence representation after memory consolidation”): "The resulting map thus represents the subset of voxels that fulfill the union of two logical conjunctions (Consolidated>New)∩(Consolidated>0))∪((New>Consolidated)∩(New>0)) with each sub-test being assessed by thresholding the p-value of individual non-parametric permutation tests."

[Editors' note: further revisions were requested prior to acceptance, as described below.]

Essential revisions:1) Concern that the statistical tests are too liberal. The contrasts appear to be global null conjunction tests. There has been significant debate regarding the usage of global null conjunction in fMRI research (see commentary on conjunctions involving papers by Nichols and Friston, 2006). Conjunctions of the global null do not require that activation for both (or however many) individual maps supplied to the conjunction to be individually significant, and as result, small effects can be amplified. Further, most often in the literature conjunctions are performed between multiple differential condition contrasts (eg., CondA>CondB with CondA>CondC), and not (CondA>CondB with CondA>baseline). The primary concern here is with the contrast involving differences between consolidated and novel sequences. The authors test: (consolidated > new) ∧ (consolidated > 0); and then for new: (new > consolidated) ∧ (new > 0). It is unclear why contrasts relative to baseline are integrated in the contrast. We recognize that there is reason to want to correct for effects that are positive, but a more parsimonious version of this would be to use a liberal mask for positive effects, using the contrast (consolidated > new), where there is no risk of inflating statistics. Instead, by incorporating the baseline contrast in the conjunction the effect is driven by (consolidated > 0), and not the more subtle effects that are of interest (e.g., consolidated > new). It is critical that the authors demonstrate the effects are present without the baseline conjunction test. Again, there is no need for a conjunction. Simple contrasts (consolidated > new, masked by consolidated > 0) and vice versa seem to be a preferable solution and we would like to see this in lieu of the conjunction analysis.

We agree with the reviewer that our use of the term conjunction may have been misleading, and that testing against global-null should be avoided. In fact, in our analyses, we did not use the global-null conjunction approach (Friston et al., 1999) but used instead the more conservative approach that was later proposed to provide nominal false-positive rate (Nichols et al., 2005) that equates to masking. Therefore, each test (contrast and simple effect) are individually significant, as assessed through permutation. We thus added the reference to Nichols et al., 2005 in the manuscript (subsection “Reorganization of the distributed sequence representation after memory consolidation”, subsection “Statistical testing”) and mentioned that it is equivalent to masking (“This procedure equals to mask the contrast with significantly positive simple effects.”, subsection “Reorganization of the distributed sequence representation after memory consolidation”; “For the contrast between consolidated and new sequences we applied a minimum-statistic conjunction test (Nichols et al. 2005) (equivalent to masking positive and negative differences with the respective simple effect) to assess that the differences found were supported by statistically significant above zero Mahalanobis distances.”, subsection “Statistical testing”) in order to clarify our use of the conjunction analysis.

2) In the retest session before the MVPA session, is there an overnight gain in performance (i.e., gain in performance measure between the last trial of previous day and the first trial of the next day) in the two trained sequences? This is the typical behavioural evidence for motor memory consolidation (e.g., Stickgold, 2005), and we think it should be incorporated here.

We agree with the reviewers that consolidation of MSL if often assessed by offline gains in performance. However, many factors can influence the appearance of these gains (Pan and Rickard, 2015). First, among these factors, the number of blocks taken into account at the end of training and beginning of retest can largely influence the amount of gains. The majority of studies that revealed such gains have used performance based upon 3 or 4 blocks in order to reduce variance, and thus counteract the transiently deteriorated performance often seen at the beginning of retest caused by the “warm-up” phenomenon. In our study, performance in the post-sleep retest session of the trained sequence was measured using one-block of 12 sequences only, and the latter was compared to the last block of training in order to compare apples with apples. Using such a design, no spontaneous gains in performance was observed. Importantly, however, no loss in performance was observed either the next day, hence suggesting evidence of consolidation. In fact, it has been theoretically argued that consolidation can be assessed by retention as opposed to the performance decay observed when no sleep is allowed (Nettersheim et al., 2015); a pattern of results that was found here in our study. Second, our results show a persistent performance advantage for the consolidated sequences compared to new ones. In our opinion, this isolates even better the sequence learning consolidation effect from other generalizable components of the task, which can be learned on the first day and transferred to the new sequence. This also allows to control for many factors (fatigue, circadian cycle, etc.) that modulates the memory and motor performances in sleep studies. Thus, while our way of measuring consolidation deviates from the common approach in the field, including in studies from our group, we think that the participants clearly showed evidence of consolidation of the trained sequences. In sum, we support the idea that the consolidation of motor memory does not requires gains but rather retention of performance level attained at the end of the training (similarly to what occurs with declarative memory), and that comparing performance of consolidated knowledge to that of new one at the same point in time does have benefit in controlling many experimental factors.

As a comparison to previous studies, we tested whether the mean correct sequence duration of the last 2 blocks of the training sessions differed from the first 2 blocks of the retest sessions on day 3 (fix-effect:pre-post, random-effect:sequence), and the result was significant (β =−0.086, SE=0.023, p=.00014).

We added the following sentences to the behavioral Results section: “While most studies in the field have measured retention and even offline gains between the end of training and the beginning of a post-sleep retest, we chose here to measure consolidation as the difference at the same point in time between the new and consolidated sequences. We also measured the difference in speed (mean correct sequence duration) between the last two blocks of training and the first two blocks of retest for the consolidated sequences (fixed-effect:post, random-effect:sequence) in order to be comparable to previous studies, and found a significantly lower sequence duration in the beginning of the retest (β =−0.086, SE=0.023, p=.00014).”

3) The consolidated sequences were not learned at the same time and not subject to the same consolidation and reconsolidation period. These differences could ultimately bias the classification results for the consolidated sequences. Tseq1 was practiced on Day1, and then tested on Day2. Tseq2 was then immediately practiced after Tseq1. Tseq1 and Tseq2 were tested in the scanner protocol on Day3. There are 2 problems. First, Tseq1 was consolidated over 2 nights of sleep, versus 1 for Tseq2. Because of this, any difference in neural representation could be due to the simple age and consolidation history of the memory. Second, exposure to a new sequence (Tseq2) immediately following the reactivation of Tseq1 can interfere with the reconsolidation of Tseq1. Evidence from recent studies of motor reconsolidation have shown that exposure to similar manipulation can significantly alter the memory trace for the reactivated, previously consolidated memory (Tseq1). Essentially what is left is a classification between a sequence that was consolidated and reconsolidated vs. a sequence that was consolidated. It is unclear why the authors did not train both of the sequences at the same time particularly because they are testing them as such. This limitation should be addressed.

The reviewers are right that this is a limitation of our study, as this was not primarily designed for the present analysis, but intended to measure consolidation and reconsolidation of motor sequence learning. However, many models posit that reconsolidation uses the same mechanism as consolidation, and the cerebral structures that support the long-term memory should not differ. Therefore, the consistent pattern differences between consolidated sequences localized in this study should reflect genuine sequence encoding. This is further supported by the absence of behavioral difference between the consolidated and reconsolidated sequences.

Accordingly, we added the following sentence to the discussion to address this limitation: “Similarly, due to the main goal of the study, which was to investigate consolidation and reconsolidation of sequential motor learning, the trained sequences were not learned on the same day, potentially inducing modulating effect on the brain patterns that were measured. However, as reconsolidation is thought to rely on mechanisms similar to consolidation, we believe that both sequences were supported by local networks in the same regions that evoked differentiated patterns of activity rather than by large-scale reconsolidation induced activity changes.” (subsection “Methodological considerations”).

4) Regarding the performance issue (i.e., improvement of new sequences during the MVPA session): Critically, in their response, the authors argued that the inconsistency of Author response image 1 to Figure 3 is due to the lack of statistical power because of using only one cross-validation fold. However, at the same time, they argued that the inconsistency of Author response image 2 (the contrast of distances between the first and the second run) to Figure 3 is the supporting evidence that the result presented in Figure 3 reflects consolidation, despite that these estimates were also computed using only one cross-validation fold. If they say Author response image 1 is unreliable because of using single fold, then what is presented in the Author response image 2 should also be unreliable for the very same reason. Ideally, the new sequences should have practiced well until their performance and underlying representations reached to plateau before going into the scanner, as the initial rapid change in the representation would essentially have little thing to do with consolidation process. It is still true that the distance estimate computed through the 4 partitions would be significantly biased if there is some inconsistency in pattern difference in any of the partitions due to the ongoing performance improvement. Therefore, caution is required to conclude that the difference highlighted in Figure 3 can be attributed specifically to the consolidation. The authors should discuss the potential impact of the sub-optimal estimate of the plateau representation for the new sequences.

We would like to thank the reviewer for noting this issue. We thus added the following sentence in the text of the manuscript to address the sub-optimal estimate of the plateau representation for new sequences: “In particular, it can be noted from Figure 1 that the subjects’ performance of the new sequences evolved as they practiced them. Thus, as we assessed differences in patterns evoked by their execution across the whole-session, it is possible that representations emerging when the performance plateaued were missed by our analysis. This problem is inherent to any investigation of learning mechanisms and should be addressed in future studies measuring the dynamic of pattern difference changes within the training session which would require both a sensitive metric and signal of highest quality.” (subsection “Methodological considerations”).